# Clinical evaluation of the SD Biosensor SARS-CoV-2 saliva antigen rapid test with symptomatic and asymptomatic, non-hospitalized patients

**Zsofia Igloi**[1]*, **Jans Velzing**[1], **Robin Huisman**[1], **Corine Geurtsvankessel**[1], **Anoushka Comvalius**[1], **Jeroen IJpelaar**[1], **Janko van Beek**[1], **Roel Ensing**[2], **Timo Boelsums**[2], **Marion Koopmans**[1], **Richard Molenkamp**[1]

1 Department of Viroscience, Erasmus Medical Centre, Rotterdam, The Netherlands, 2 Public Health Service Rotterdam-Rijnmond, Rotterdam, The Netherlands

* z.igloi@erasmusmc.nl

**Data Availability Statement:** All relevant data are within the manuscript and its Supporting Information files.

## Abstract

### Background

Performance of the SD Biosensor saliva antigen rapid test was evaluated at a large designated testing site in non-hospitalized patients, with or without symptoms.

### Method

All eligible people over 18 years of age presenting for a booked appointment at the designated SARS-CoV-2 testing site were approached for inclusion and enrolled following verbal informed consent. One nasopharyngeal swab was taken to carry out the default antigen rapid test from which the results were reported back to the patient and one saliva sample was self-taken according to verbal instruction on site. This was used for the saliva antigen rapid test, the RT-PCR and for virus culture. Sensitivity of the saliva antigen rapid test was analyzed in two ways: i, compared to saliva RT-PCR; and ii, compared to virus culture of the saliva samples. Study participants were also asked to fill in a short questionnaire stating age, sex, date of symptom onset. Recommended time of ≥30mins since last meal, drink or cigarette if applicable was also recorded. The study was carried out in February-March 2021 for 4 weeks.

### Results

We could include 789 people with complete records and results. Compared to saliva RT-PCR, overall sensitivity and specificity of the saliva antigen rapid test was 66.1% and 99.6% which increased to 88.6% with Ct ≤30 cutoff. Analysis by days post onset did not result in higher sensitivities because the large majority of people were in the very early phase of disease ie <3 days post onset. When breaking down the data for symptomatic and asymptomatic individuals, sensitivity ranged from 69.2% to 50% respectively, however the total

**Funding:** The SARS-CoV-2 Rapid Antigen Test-Standard Q COVID-19 Ag Saliva- Research use only (Lot number QCO9021001; expiry date 04-01-2023) was provided by SD Biosensor. The funders had no role in study design, data collection and analysis, decision to publish, or preparation of the manuscript.

**Competing interests:** The authors have declared that no competing interests exist.

number of RT-PCR positive asymptomatic participants was very low (n = 5). Importantly, almost all culture positive samples were detected by the rapid test.

## Conclusion

Overall, the potential benefits of saliva antigen rapid test, could outweigh the lower sensitivity compared to nasopharyngeal antigen rapid test in a comprehensive testing strategy, especially for home/self-testing and in vulnerable populations like elderly, disabled or children where in intrusive testing is either not possible or causes unnecessary stress.

## Introduction

In spite of having multiple vaccines available [1], the COVID-19 pandemic is far from over and testing remains an essential pillar of the pandemic response. In high income countries, by now sufficient diagnostic capacities have been established however access to testing can be hindered by remote locations, lack of information, financial obstacles etc. Besides access, willingness is another factor which can be increased by simple, comfortable and non-invasive testing options. To increase access and testing rates, self / home testing approaches are explored and being implemented worldwide. However there is considerable debate on the drawbacks mostly from the disease surveillance and reliability of the results point of view [2]. This is because the multiple components which cannot be controlled over time when a non-trained professional carries out a test. The first step to make self / home testing an acceptable option is the availability of good reliable tests which are CE marked and fully evaluated for this specific application; these are still scarcely available [3]. Furthermore currently besides a few [4–6], most antigen rapid detection tests (RDTs) have only been validated using nasopharyngeal swabs which arguably can best be executed by trained health-care professionals and are less suited for self-testing. Suitability of saliva/oral fluid has been explored as a sample type both for molecular and serological test and found both higher and lower sensitivity for saliva samples compared with nasopharyngeal swabs. However, meta-analyses of such studies suggest an overall similar or non-statistically significant lower sensitivity associated with the use of saliva samples [7]. Taking patient comfort and ease of sampling in account saliva is definitely well suited also for self-sampling.

Because of the above mentioned reasons, we evaluated the clinical performance of a RDT utilizing saliva as sample amongst both symptomatic and asymptomatic general population presenting at testing locations. In order to have reliable data of the performance, samples were collected and tested by trained professionals. We have found acceptable [8] performance among symptomatic individuals who were in the early onset ie <7 days of disease. Results from this study can encourage further evaluation of RDTs using saliva thereby laying down the road towards inclusion saliva RDT in self-testing approaches.

## Materials and methods

### Testing site, testing procedures, population and patient recruitment

The study was carried out at an XL testing location in Rotterdam—Rijnmond (Rotterdam is the second largest city in the Netherlands) which is by appointment only. Persons with any respiratory symptoms, or persons that have been contacts of confirmed cases regardless of symptoms are eligible for a free of charge SARS-CoV-2 test. Vulnerable persons (ie elderly or chronic

condition) and priority groups (ie teachers, healthcare workers) are tested by reverse transcrip-tase polymerase chain reaction (RT-PCR), everyone else is tested by the SARS-CoV-2 Rapid Antigen Test (Distributed by Roche (SD Biosensor)) using a nasopharyngeal (NP) swab. In this study persons eligible for a rapid antigen test were enrolled. The SARS-CoV-2 Rapid Antigen Test-Standard Q COVID-19 Ag Saliva- Research use only (Lot number QCO9021001; expiry date 04-01-2023) was provided by SD Biosensor (http://www.sdbiosensor.com/xe/) but at time of writing this rapid antigen test was not available on the market.

At the entrance of the testing site all eligible people over 18 years of age were approached for inclusion and enrolled following verbal informed consent. Study participants were also asked to fill in a questionnaire stating age, sex, date of symptom onset. Recommended time of ≥30mins since last meal, drink or cigarette if applicable was also recorded. The study was car-ried out for 4 weeks (10 February-19 March) to reach an ideally total of 1000 inclusions. Study was terminated just prior to reaching the target at 816 inclusions for logistic reasons. From this, 789 had a PCR result due to a laboratory logistic problems.

## Specimen collection, testing and culture procedures

Saliva was collected based on instructions of use by the test provider (nasal discharge and pos-terior pharyngeal spitting drooled into collection device) using the Zeesan Saliva RNA collec-tion kit without preservation medium (http://www.zeesandx.com/coronavirus/1075.html). All saliva samples were tested for SARS-CoV-2 RNA using the cobas ® SARS-CoV-2 RT-PCR test on the COBAS6800 (Roche diagnostics). Genome copies/ml were calculated based on an in house established standard curve. All RT-PCR positive saliva samples were inoculated onto Vero cells following dilution with universal transport media (VTM) (HiViral; HiMedia Labo-ratories PVT, Ltd., https://www.himedialabs.com) to 6ml volume, filtration with 45μm bacte-rial filter and mixing with FBS: 1080ul saliva+VTM with 720ul FBS prior to freezing at -80˚C for a maximum of two weeks. Samples were cultured for a maximum of 14 days or until cyto-pathic effect (CPE) was visible. The presence of SARS-CoV-2 was confirmed by immunofluo-rescence, using a rabbit polyclonal antibody targeting SARS CoV-2 nucleocapsid protein (Sino Biological Inc.).

## Data analysis

Data from the RDTs, RT-PCR, virus culture and clinical questionnaire were merged using Microsoft Access (http://www.microsoft.com), and data analysis was performed using Micro-soft excel and R software version 4.0.2 (https://www.r-project.org). Sensitivity and specificity of RDTs were calculated in relation to the saliva RT-PCR and virus culture results. As a com-parator sensitivity of the nasopharyngeal RDT was also calculated however only to saliva RT-PCR results as no PCR could be carried out on nasopharyngeal samples. Negative and pos-itive predictive values (NPV and PPV) were calculated using percentage PCR positivity figures as a good proxy for disease prevalence. Clopper-Pearson analysis was used to determine confi-dence intervals of proportions. Two sample t-test was used to define significance of difference between means.

## Ethical clearance

The medical research ethics committee (MREC) of Erasmus Medical center decided the study was not subject to the Medical Research Involving Human Subjects Act (WMO) and did not require full review by an accredited MREC (protocol number MEC-2021-0083).

## Results

In total 789 complete dataset were available for analysis. Results of both the saliva and the nasopharyngeal RDT and the virus culture was compared to RT-PCR results and categorized by RT- PCR Ct values as an indicator for viral load and detection limit, Table 1. Participants had a median age of 37 years, an equal proportion of males and female (male, 50.6%) and almost a quarter were smokers. Most people presenting for testing had symptoms (70.5%, 556/789) with recent onset of disease (median 2 days post onset). Of the symptomatic people who could provide the exact date of onset (545/556) vast majority was in the very early phase of disease (77.5% <3days) which is also seen in the proportion of low Ct values amongst the study population (70% Ct≤30). Of all participants, 7.9% tested positive by RT-PCR what was just below the current percentage positivity nationally (between 11.2%-8.6% during the course of

**Table 1. Characteristics of the study population.**

| | |
|---|---|
| **Total N (RT-PCR and RDT results)** | 789 |
| **Age [median (min-max); N]** | 37 years (18–79 years) |
| **Sex [%M, (n/N)]** | 50.6% (399/789) |
| **Smoker [%Y, (n/N)]** | 20.8 (164/789) |
| **Symptoms present [%Y, (n/N)]** | 70.5% (556/789) |
| **Days from symptom onset [median (min-max); N]** | 2 days (0–41); 545 |
| **Days 0–3 [n/N (%)]** | 431/545 (77.5%) |
| **Days 4–7 [n/N (%)]** | 91/545 (16.4%) |
| **Days 8+ [n/N (%)]** | 23/545 (4.1%) |
| **Positivity PCR SARS-2 E gene [%, (n/N)]** | 7.9% (62/789) |
| **PCR Ct [median (min-max); N]** | 27.6 (17.4–35.1); 62 |
| **Ct > 30 [n, (%)]** | 18 29% |
| **Ct ≤ 30 [n, (%)]** | 44, 70.1% |
| **Ct ≤ 25 [n, (%)]** | 25, 40.3% |
| **Positivity NP RDT [%, (n/N)]** | 6.6% (52/789) |
| **NP RDT samples with positive PCR result [n]** | 49 |
| **PCR Ct SARS-2 E gene [median (min-max); N]** | 26.6 (17.4–34.2); 49 |
| **Ct > 30 [n, (%)]** | 9, 18.4% |
| **Ct ≤ 30 [n, (%)]** | 40, 81.6% |
| **Ct ≤ 25 [n, (%)]** | 22, 44.9% |
| **Positivity saliva RDT [%, (n/N)]** | 5.6% (44/789) |
| **Saliva RDT samples with positive PCR result [n]** | 41 |
| **PCR Ct [median (min-max); N]** | 25.5 (17.4–34.2); 41 |
| **Ct > 30 [n, (%)]** | 2, 4.9% |
| **Ct ≤ 30 [n, (%)]** | 39, 95.1% |
| **Ct ≤ 25 [n, (%)]** | 25, 60.9% |
| **Positivity saliva virus culture [%, (n/N)]** | 48.4% (30/62) |
| **Ct > 30 [n, (%)]** | 0, 0% |
| **Ct ≤ 30 [n, (%)]** | 30, 100% |
| **Ct ≤ 25 [n, (%)]** | 21, 70% |

Data of all included people in the study were analyzed by basic demographics, smoking status, date of disease onset, RT-PCR Ct values and virus culture result.

RT-PCR, reverse transcription PCR; RDT, antigen rapid detection test; Min-max, minimum and maximum; M, male; Y, yes; n/N, amount of total sample size NP, nasopharyngeal swab; Ct, cycle threshold; E gene, envelope gene;

the study) [9]. More positive samples were detected by the standard nasopharyngeal RDT than by saliva RDT (n = 52 vs. n = 44). All samples with Ct≤30 could be cultured.

Sensitivity and specificity of the saliva RDT was analyzed in two ways: i, compared to saliva RT-PCR Table 2 and Fig 1; and ii, compared to virus culture Fig 1. Comparing to RT-PCR in saliva, overall sensitivity of the saliva RDT was 66.1%. When PCR Ct ≤30 (E gene copy/ml 2.17E+05) cutoff was used sensitivity increased to 88.6%. When breaking down the data for symptomatic and asymptomatic individuals, sensitivity ranged from 69.2% to 50% respectively, however the total number of RT-PCR positive asymptomatic participants was very low (n = 5). Analysis by days post onset did not result in higher sensitivities but this is due to the uneven representation of patients with various days post onset in the study population (majority <3 days). Specificity was comparable to previous findings using nasopharyngeal RDT compared to PCR (between 99.1% and 99.8%) [10].

As nasopharyngeal RDT was used for standard diagnostic in this population we have also calculated sensitivity and specificity of the nasopharyngeal RDT compared to RT-PCR in saliva as the only available material and compared the performance of the two RDTs. Overall sensitivity of saliva RDT was lower (66.1% vs. 79.0%). Similarly to saliva RDT, analysis by days post onset did not result in higher sensitivities but this is due to the uneven representation of patients with various days post onset in the study population (majority <3 days). Analysis by

**Table 2. Sensitivity and specificity of both saliva and nasopharyngeal antigen RDT compared to RT-PCR.**

| Saliva RDT | Compared to saliva RT-PCR | | | | |
|---|---|---|---|---|---|
| | 0–3 days post onset | 0–7 days post onset | Overall | Symptomatic | Asymptomatic |
| Sensitivity% (CI95%), n | 68.5% | 65.0% | 66.1% | 69.2% | 50.0% |
| | (54.5–80.5) (37) | (51.6–76.9) (39) | (52.9–77.6) (41) | (54.9–81.3) (36) | (18.7–81.3) (5) |
| ≤Ct 30, Sensitivity% (CI95%), n | 87.5% | 88.1% | 88.6% | 88.2% | 83.3% |
| | (73.2–95.8) (35) | (74.4–96.0) (37) | (75.4–96.2) (39) | (72.6–96.7) (30) | (35.9–99.6) (5) |
| Specificity% (CI95%), n | 99.7% | 99.6% | 99.6% | 99.8% | 99.1% |
| | (98.8–99.9) (619) | (98.8–99.9) (703) | (98.8–99.9) (724) | (98.9–99.9) (503) | (96.8–9.9) (221) |
| NPV% (CI95%) | 96.7% | 96.3% | 96.4% | 96.7% | 94.8% |
| | (95.1–97.7) | (94.8–97.3) | (95.0–97.4) | (95.2–97.8) | (90.7–97.1) |
| PPV% (CI95%) | 95.9% | 96.3% | 94.6% | 97.5% | 86.0% |
| | (85.3–99.0) | (94.8–97.3) | (84.9–98.2) | (84.3–99.7) | (57.4–96.5) |
| NP RDT | Compared to saliva RT-PCR | | | | |
| | 0–3 days post onset | 0–7 days post onset | Overall | Symptomatic | Asymptomatic |
| Sensitivity% (CI95%), n | 79.2% | 78.7% | 79.0% | 82.7% | 60.0% |
| | (65.0–89.5) (38) | (66.3–88.1) (48) | (66.8–88.3)(49) | (69.7–91.8) (43) | (26.2–87.9) (6) |
| ≤Ct 30, Sensitivity% (CI95%), n | 90.0% | 90.5% | 90.9% | 91.2% | 83.3% |
| | (76.3–97.2) (36) | (77.4–97.3) (38) | (78.3–97.5) (40) | (76.3–98.1) (31) | (35.9–99.6) (5) |
| Specificity% (CI95%), n | 99.5% | 99.6% | 99.6% | 99.6% | 99.6% |
| | (98.6–99.9) (618) | (98.8–99.9) (703) | (98.8–99.9) (724) | (98.6–99.9) (502) | (97.5–99.9) (222) |
| NPV% (CI95%) | 97.8% | 97.7% | 97.8% | 98.1% | 95.8% |
| | (96.2–98.7) | (96.3–98.6) | (96.4–98.6) | (96.7–99.0) | (91.4–98.0) |
| PPV% (CI95%) | 94.7% | 95.3% | 95.5% | 95.8% | 93.6% |
| | (85.2–98.3) | (86.7–98.5) | (87.1–98.5) | (85.1–98.9) | (66.1–99.1) |

Overall and stratified sensitivity and specificity with 95% confidence interval (CI95%) of the saliva and nasopharyngeal antigen RDT compared to RT PCR. Results were also analyzed by days post onset and symptom status. Negative and positive predictive values (NPV and PPV) were calculated using the mean 9.9% PCR positivity nationally as a proxy for prevalence [9] during the study period.

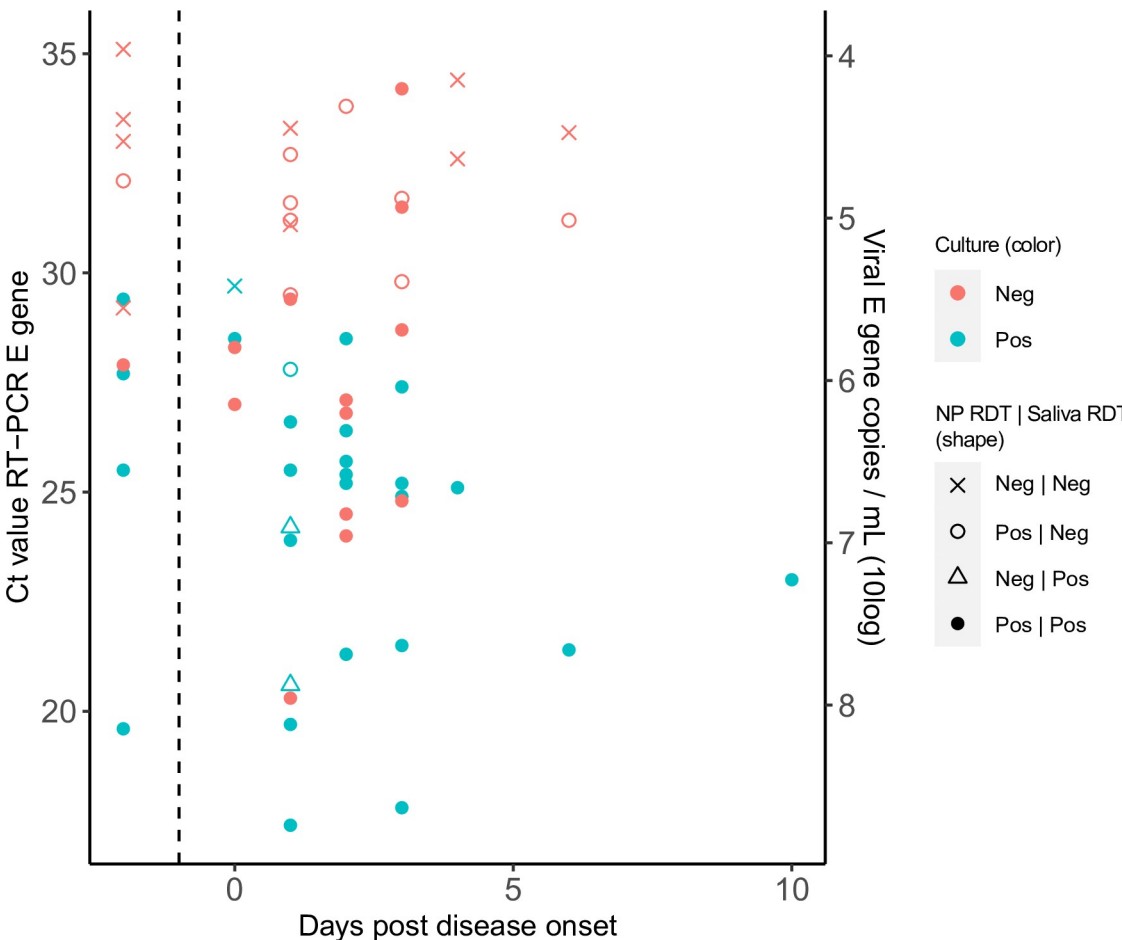

**Fig 1. Cycle thresholds and genome copies of RT-PCR positive samples in relation to days since symptom onset, saliva and nasopharyngeal RDT positivity, and culture outcomes of participation with both symptomatic and asymptomatic patient (n = 62).** Data points shown on the left side of the dashed bar are from asymptomatic individuals. NP, nasopharyngeal swab; RDT, antigen rapid detection test; $C_t$, cycle threshold; E gene, envelope gene; Neg, negative; Pos, positive; RT-PCR, reverse transcription PCR.

symptom status or by days post onset separately a similar difference was seen (symptoms present 69.2% vs. 82.7; <3 days 68.5% vs 79.2%). However sensitivity was similar with Ct ≤30 cut-off (88.6% vs. 90.9%). Generally sensitivity of nasopharyngeal RDT was lower but specificity was equally high and comparable to our previous study [10].

All PCR positive saliva samples were inoculated onto cell culture and approximately half (48.4%, 30/62) resulted in a positive CPE (mean Ct 24.2 of positive samples) Fig 1. Almost all samples with a positive culture result were detected either by nasopharyngeal RDT (90%, 27/30) or by saliva RDT (97%, 29/30). Only one sample was not detected by both RDTs and demonstrated a relatively high Ct-value (Ct 29). Culture negative samples (51.6%, 32/62) had higher Ct values (mean Ct 30.2 (p = 0.001). Results of the two RDTs were partially in concordance as 10/32 were not detected by neither tests. The n = 9 culture negative samples which were not detected by the saliva RDT were in the high Ct value range (median Ct 31.5). However there were n = 13 RDT positive but culture negative samples ranging from low to high Ct.

## Discussion

There is very limited data available on the performance of saliva based RDTs [11, 12] and those that are available show below optimal results. The published studies report sensitivities of 63% and 24% compared to nasopharyngeal or nasal swab utilizing RDTs. In our study we have found that people with high viral load (Ct≤30 cutoff) could be detected with a saliva antigen RDT with relatively high sensitivity (88.6%). Furthermore, the majority of presumed infectious individuals (based on cell culture positivity in saliva) could be detected (96.7%). However there is still much to learn about viral kinetics in saliva and how it compares in patients with various disease severity, to other respiratory samples and the role of SARS-CoV-2 specific antibodies in saliva.

Currently nasopharyngeal swab remain the gold standard method for both PCR and RDT with room for exploration for saliva as evidence is still mostly lacking to fully understand the usability and reliability at various time points during disease and in different populations over time [7]. In this study, nasopharyngeal RDT was used as default diagnostic method and also as an additional reference method for the saliva RDT and demonstrated lower sensitivity and comparable specificity to our previous evaluation [10]. Reasons for this difference can be the changing population and also the comparator which was saliva RT-PCR. Using saliva RDT, despite lower general sensitivity compared to both saliva RT-PCR and nasopharyngeal RDT, based on RT-PCR Ct/viral load values almost all presumed infectious individuals could be detected, however this result could not be compared to nasopharyngeal swab virus culture in the current study. Saliva is a complex material shown to have comparable or slightly lower viral load to nasopharyngeal sample [13], it also contains SARS-2 specific antibodies [14] and different viral kinetics compared to nasopharynx which might explain the lower performance in the tested population. Furthermore, we have evaluated this test on non-hospitalized patients and a study found correlation of more severe disease with higher viral load in saliva [15].

Although our study is limited in the suboptimal number of positive samples, it provides promise for further exploration of saliva based RDTs to be used also for self/home testing. We did not investigate acceptance of saliva vs other invasive sample types but one could argue that collection of saliva is easier to perform and causes minimal discomfort and some studies showed high acceptability [16]. Limited studies on usability of RDTs showed relatively high sensitivity and low false negativity related to self-testing and these studies used either nasal or nasopharyngeal samples [17]. However comprehensive studies are still mostly lacking. Risk of aerosol generation could be higher in oral fluid sampling depending on exact sample type involving forced coughing. This risk for home /self–testing does not present an issue however for testing sites proper mitigation steps and set-up need to be thought of.

Availability of easy to use, comfortable RDTs might also increase willingness to test. Multiple studies showed that ease of access ie by central location of testing site or positive experience by minimizing discomfort and rapid availability of results all increased willingness [2, 18, 19]. Increased testing also helps with prompt identification and isolation of cases thereby stopping the spread. However the reliability of the test result depends also on disease prevalence in the population when the test is taken. At the time of our study the mean disease prevalence was 30/100.000 with a mean % PCR positivity of 9.9% resulting in a positive predictive value of the saliva RDT of 94.6%. By the time of the writing of the manuscript the lowest and the highest prevalence ranged between 2.9 (2.3% PCR positivity) and 58.4/100.00 (20.1% PCR positivity). If we apply these scenarios for the test performance, this would translate into 79.1%-97.6% PPV. Given the good specificity of the test the even under high prevalence the NPV still remains over 90% (99.2%– 92.1%) but there will still be individuals who obtain false negative results. The impact on transmission of false negative results should be considered, as

individuals may demonstrate lower adherence to non-pharmaceutical measures believing that they tested negative. Furthermore with already high and still increasing vaccination rates continuous decline in prevalence could be expected and need to be taken into account for future testing policies.

Overall, the potential benefits of saliva RDT, could outweigh the lower sensitivity in a targeted and comprehensive testing strategy, especially for home/self-testing and in vulnerable populations like elderly, disabled or children where in intrusive testing is either not possible or causes unnecessary stress.

## Supporting information

**S1 File.**
(XLSX)

## Acknowledgments

Testing was carried out at the Ahoy XL testing centre in Rotterdam, the Netherlands, where numerous people contributed to the success of this project. We would like to thank all study participants, employees of the GGD Rotterdam—Rijmond especially all laboratory coordinators, namely Jelena Pajic, Lars Kortenhorst, Niekeline Kimmel, Ono Bestman, Bert de Valk, Sylviana Anthony for participation, coordination and the laboratory workers who carried out the rapid antigen tests.

## Author Contributions

**Conceptualization:** Zsofia Igloi, Jans Velzing, Robin Huisman, Corine Geurtsvankessel, Roel Ensing, Timo Boelsums, Richard Molenkamp.

**Data curation:** Robin Huisman.

**Formal analysis:** Zsofia Igloi, Janko van Beek.

**Investigation:** Zsofia Igloi.

**Methodology:** Zsofia Igloi, Robin Huisman, Corine Geurtsvankessel, Anoushka Comvalius, Jeroen IJpelaar, Richard Molenkamp.

**Supervision:** Zsofia Igloi, Jans Velzing, Richard Molenkamp.

**Writing – original draft:** Zsofia Igloi.

**Writing – review & editing:** Corine Geurtsvankessel, Anoushka Comvalius, Janko van Beek, Marion Koopmans, Richard Molenkamp.

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
