## [Decision Letter · Decision Letter 0]

17 Sep 2021

PONE-D-21-21141Clinical evaluation of the SD Biosensor saliva antigen rapid test with symptomatic and asymptomatic, non-hospitalized patients.PLOS ONE

Dear Dr. Iglói,

Thank you for submitting your manuscript to PLOS ONE. After careful consideration, we feel that it has merit but does not fully meet PLOS ONE’s publication criteria as it currently stands. Therefore, we invite you to submit a revised version of the manuscript that addresses the points raised during the review process.

 Please re-write the results in detail.

We look forward to receiving your revised manuscript.

Kind regards,

Etsuro Ito

Academic Editor

PLOS ONE

Journal Requirements:

NO - Include this sentence at the end of your statement: The funders had no role in study design, data collection and analysis, decision to publish, or preparation of the manuscript.

3. Thank you for stating the following in the Acknowledgments/ Funding Section of your manuscript: 

The SARS-CoV-2 Rapid Antigen Test-Standard Q COVID-19 Ag Saliva- Research use only (Lot number QCO9021001; expiry date 04-01-2023) was provided by SD Biosensor 

No other funding was received.

NO - Include this sentence at the end of your statement: The funders had no role in study design, data collection and analysis, decision to publish, or preparation of the manuscript.

4. Please upload a new copy of Figure 1 as the format is not accepted. Please follow the link for more information: https://blogs.plos.org/plos/2019/06/looking-good-tips-for-creating-your-plos-figures-graphics/" https://blogs.plos.org/plos/2019/06/looking-good-tips-for-creating-your-plos-figures-graphics/

Reviewers' comments:

Reviewer's Responses to Questions

**Comments to the Author**

1. Is the manuscript technically sound, and do the data support the conclusions?

Reviewer #1: Partly

Reviewer #2: Yes

2. Has the statistical analysis been performed appropriately and rigorously? 

Reviewer #1: Yes

Reviewer #2: No

3. Have the authors made all data underlying the findings in their manuscript fully available?

Reviewer #1: Yes

Reviewer #2: Yes

4. Is the manuscript presented in an intelligible fashion and written in standard English?

Reviewer #1: Yes

Reviewer #2: Yes

5. Review Comments to the Author

Reviewer #1: The manuscript is clearly written and presents public health relevant findings. It will benefit from minor revision in relation to the data presentation and interpretation in the public health context.

Title: it would be helpful to specify in the title that this saliva test is for SARS-CoV2 antigen.

Abstract: clear and appropriate

Introduction: It would be valuable to introduce an applied epidemiology perspective here; such as

a) The manuscript discusses PPV, yet in public health, the NPV can be equally valuable. Especially in the context of self-tests being used by citizens prior to engaging in increased social interaction. Please elaborate on this perspective (useful reference: European Centre for Disease Prevention and Control. Considerations on the use of self-tests for COVID-19 in the EU/EEA – 17 March 2021. ECDC: Stockholm; 2021)

b) the dependency of population prevalence. In the manuscript, only the prevalence among the tested population is used, though it would be insightful to understand what was the population COVID19 prevalence at the time of this study;

c) Please describe what is known about access to testing in the described population of Rotterdam: how large is the estimated group that is unable to come to these XL-locations? This is relevant context in the discussion on the value of self tests.

Methods: Clearly written. It would be helpful to clarify why only 789 of the 816 included persons had a PCR result.

Results:

- Define 'recent onset' (P4-L89) (from table 1, we deduce this must be <8 days, but this could be made explicit)

- table 1:

Every time when "CT<=30" is printed in the table, this should be "CT<30"

The numbers of CT>=30 and CT<30 among the NP AgRDT group should add up, yet they don't: 9+39=48, yet 49 (of 52) should have a positive PCR. Please check the data

- Table 2: (page 7)

It is not clear why the N is larger than 556 in the columns of "0-3 days post onset" and "0-7 days post onset". It is my understanding that only 556 participants indicated an onset of symptoms.

The row <ct30 91.2="" a="" among="" bit="" can="" clinical="" confusing:="" explain="" group="" how="" is="" overall="" sensitivity="" symptomatic="" that="" the="" we="">

- Table 2 (page 8)

Here too: It is not clear why the N is larger than 556 in the columns of "0-3 days post onset" and "0-7 days post onset". It is my understanding that only 556 participants indicated an onset of symptoms.

Same on page 9

Discussion:

Clear focus chosen on the discussion topics. What is missing, is a perspective on public health consequences of the PPV and the NPV found in this study: it would not be inappropriate to indulge in a 'what if' scenario, exploring how the saliva test would perform if used widely in the given population of Rotterdam, given the estimated COVID19 prevalence at that time.

Lastly, it would be of interest to elaborate a bit on the risk of aerosolisation while taking the saliva samples, and discuss how that would compare to the other tests used in this study.

The numbers of CT>=30 and CT<30 among the saliva AgRDT group should add up, yet they don't: 2+38=40, yet 41 (of 44) should have a positive PCR. Please check the data</ct30>

Reviewer #2: Review of “Clinical evaluation of the SD Biosensor saliva antigen rapid test with symptomatic andasymptomatic, non-hospitalized patients”. Short article with promising results. Methods section is very clear. The results could be better displayed as far as I am concerned. Also, the statistical comparison that is made to compare sensitivity estimates by time since symptom onset needs to be looked at more closely. Furthermore, I found it very encouraging that the saliva antigen test was able to detect 97% of the positive viral cultures. This is in my opinion a very convincing result to implement this easy-to-use test as a public health tool for in particular health care workers and for the general public for SARS-CoV-2 and potentially beyond for other infectious diseases as well for example influenza.

Major points:

I find the results section too brief. I would like to see clearer sentences where it is made explicit which two tests are being compared and show how the sensitivity was estimated. For example, "of the XX samples that tested positive with PCR saliva, XX tested positive with the saliva antigen test (XX/ XX = XX sensitivity)." I find it difficult at this point to determine from the text which tests were compared and how sensitivity was estimated.

To me the main finding of this study is that the saliva RDT detected all but one positive virus cultures. Assuming virus culture is the best indicator of infectiousness, the saliva RDT is in this study the best test to detect infectious persons, and would therefore be very suitable for at home use, and therefore be an excellent public health tool to stop transmission. I think this finding deserves a better location in the results and discussion section.

Line 100: Overall performance of saliva was inferior to these other two tests. But comparing one test to another will always result in an inferior test, right? There is always going to be one sample, which was tested positive in the bench mark test, that will be negative in the evaluated test? Also, inferior in finding what? RNA or samples with infectious potential? The saliva test found more of the samples that tested positive in viral culture than the nasopharyngeal test, right? In that sense, the saliva test is superior over the nasopharyngeal test finding samples that positive with positive culture.

Table 2: I am confused about the N-values. In total, there were 789 samples tested. Of these, 62 were PCR-positive, 52 with the NP Ag RDT, and 44 with the saliva Ag RDT. The sensitivity estimates can therefore never be estimated with the numbers I am seeing in this table. I would provide the actual numbers from which the sensitivity estimates were estimates in the parentheses instead of one N number, for example: 79.2% (XX/XX)

“Analysis by days post onset did not result in higher sensitivities”. How was this concluded? Looking at table 2, I see percentages for the group 0-3 days post onset and the group 0-7 post onset. For a proper comparison, one would have to make two different groups, for example comparing samples taken 0-3 days after symptom onset, and samples taken 4-7 days post symptom onset.

Minor points:

Line 47: People were either tested because of respiratory symptoms or because of being a contact of an individual with a confirmed SARS-CoV-2 infection as mentioned in line 47/48. Could you provide information on the number of samples that were from contact persons, perhaps in Table 1?

Line 62: This sentence requires rewriting.

Line 98: To me, this summary statistic is redundant and without meaning.

Line 99: Sensitivity increased to 88.6, but where is it increasing from?

6. PLOS authors have the option to publish the peer review history of their article (what does this mean?). If published, this will include your full peer review and any attached files.

Reviewer #1: **Yes: **Arnold Bosman

Reviewer #2: **Yes: **Tom Woudenberg

---

## [Author Response · Author response to Decision Letter 0]

1 Nov 2021

POINT-TO-POINT RESPONSE TO THE REVIEWER

We thank the reviewers for their valuable comments, which we will address point-to-point below. The manuscript underwent significant revision thanks to the comments. The line numbers below correspond to the manuscript without track changes.

Reviewers comments:

Reviewer #1: 

The manuscript is clearly written and presents public health relevant findings. It will benefit from minor revision in relation to the data presentation and interpretation in the public health context.

Title: it would be helpful to specify in the title that this saliva test is for SARS-CoV2 antigen.

Specified

Abstract: clear and appropriate

Introduction: It would be valuable to introduce an applied epidemiology perspective here; such as

a) The manuscript discusses PPV, yet in public health, the NPV can be equally valuable. Especially in the context of self-tests being used by citizens prior to engaging in increased social interaction. Please elaborate on this perspective (useful reference: European Centre for Disease Prevention and Control. Considerations on the use of self-tests for COVID-19 in the EU/EEA – 17 March 2021. ECDC: Stockholm; 2021)

Comment: addressed in detail in results (Table 2) and discussion. Also as a study was running for 4 weeks and during that period the amount of positive people ie % PCR positivity changed the mean value was used to calculate NPV and PPV. Furthermore in the discussion scenarios with lowest and highest % PCR positivity figures were explored. (Lines 221-231) 

b) the dependency of population prevalence. In the manuscript, only the prevalence among the tested population is used, though it would be insightful to understand what was the population COVID19 prevalence at the time of this study;

Clarified in results section (Lines 128-129)

c) Please describe what is known about access to testing in the described population of Rotterdam: how large is the estimated group that is unable to come to these XL-locations? This is relevant context in the discussion on the value of self tests.

Comment: it is difficult to know precisely, we can only estimate from the number of people having low socio economic status (SES) in Rotterdam. Several studies have shown that marginalised people often represent hard to reach population. Furthermore, self-testing is not only to reach marginalized people but also to encourage more frequent testing in any population. Some of these discussion points are included in the discussion.

Methods: Clearly written. It would be helpful to clarify why only 789 of the 816 included persons had a PCR result.

Clarified in Lines 84-85

Results:

- Define 'recent onset' (P4-L89) (from table 1, we deduce this must be <8 days, but this could be made explicit)

Comment: in this particular case (now line 125) recent onset is clarified in the bracket following the statement ie 2 days. However point taken and statement clarified throughout the manuscript. 

- table 1:

Every time when "CT<=30" is printed in the table, this should be "CT<30"

Corrected: Categories changed slightly: Ct ≤ 25, Ct ≤ 30 and Ct >30

The numbers of CT>=30 and CT<30 among the NP AgRDT group should add up, yet they don't: 9+39=48, yet 49 (of 52) should have a positive PCR. Please check the data

Correction and comment: the categories were changes as detailed in the above point. The reason for the difference in total RDT positives and the total which has ct value is due to specificity of the RDT ie false positives detected by RDT. For clarification an extra row was added in the table.

 The numbers of CT>=30 and CT<30 among the saliva AgRDT group should add up, yet they don't: 2+38=40, yet 41 (of 44) should have a positive PCR. Please check the data

Correction and comment: see comment above.

- Table 2: (page 7)

It is not clear why the N is larger than 556 in the columns of "0-3 days post onset" and "0-7 days post onset". It is my understanding that only 556 participants indicated an onset of symptoms.

Corrected and comment: indeed confusingly the total population size was displayed not the concerning true positives/true negatives which are necessary for calculating the sensitivity/specificity. 

The row 

- Table 2 (page 8)

Here too: It is not clear why the N is larger than 556 in the columns of "0-3 days post onset" and "0-7 days post onset". It is my understanding that only 556 participants indicated an onset of symptoms.

Same on page 9

Clarified: However 556 people stated presence of symptoms only 545 provided date of onset. This was clarified in the text (Line 126)

Discussion:

Clear focus chosen on the discussion topics. What is missing, is a perspective on public health consequences of the PPV and the NPV found in this study: it would not be inappropriate to indulge in a 'what if' scenario, exploring how the saliva test would perform if used widely in the given population of Rotterdam, given the estimated COVID19 prevalence at that time.

Added: Lines 221-231

Lastly, it would be of interest to elaborate a bit on the risk of aerosolisation while taking the saliva samples, and discuss how that would compare to the other tests used in this study.

Edited: the above mentioned points were addressed as much as possible in the discussion (Lines 212-215).

Reviewer #2: 

Review of “Clinical evaluation of the SD Biosensor saliva antigen rapid test with symptomatic and asymptomatic, non-hospitalized patients”. Short article with promising results. Methods section is very clear. The results could be better displayed as far as I am concerned. Also, the statistical comparison that is made to compare sensitivity estimates by time since symptom onset needs to be looked at more closely. Furthermore, I found it very encouraging that the saliva antigen test was able to detect 97% of the positive viral cultures. This is in my opinion a very convincing result to implement this easy-to-use test as a public health tool for in particular health care workers and for the general public for SARS-CoV-2 and potentially beyond for other infectious diseases as well for example influenza.

Major points:

I find the results section too brief. I would like to see clearer sentences where it is made explicit which two tests are being compared and show how the sensitivity was estimated. For example, "of the XX samples that tested positive with PCR saliva, XX tested positive with the saliva antigen test (XX/ XX = XX sensitivity)." I find it difficult at this point to determine from the text which tests were compared and how sensitivity was estimated.

Result section was rewritten and restructured and part of Table 2 removed as I believe that was confusing and redundant information.

To me the main finding of this study is that the saliva RDT detected all but one positive virus cultures. Assuming virus culture is the best indicator of infectiousness, the saliva RDT is in this study the best test to detect infectious persons, and would therefore be very suitable for at home use, and therefore be an excellent public health tool to stop transmission. I think this finding deserves a better location in the results and discussion section.

Comment: this was highlighted better both in results and discussion. In this study we did not culture from nasopharyngeal swabs however we did that in our previous studies and similar results were achieved meaning by Ct cut-off and days post onset ie early phase more infectious. In this current study days post onset did not result in increase due to the above detailed skewed dataset.

Line 100: Overall performance of saliva was inferior to these other two tests. But comparing one test to another will always result in an inferior test, right? There is always going to be one sample, which was tested positive in the bench mark test, that will be negative in the evaluated test? Also, inferior in finding what? RNA or samples with infectious potential? The saliva test found more of the samples that tested positive in viral culture than the nasopharyngeal test, right? In that sense, the saliva test is superior over the nasopharyngeal test finding samples that positive with positive culture.

Comment: Indeed most comparisons going to be inferior and in this case well expected as we compared a very sensitive molecular test to a serological test which never can be as sensitive as the molecular. It was inferior in performance to detect every RT-PCR or nasopharyngeal RDT positive person but the reviewer is right pointing out the additional value of detecting infectious individuals. Real value does come from detecting these people. Section was rephrased to reflect this (Lines 165-173 and 195-200).

Table 2: I am confused about the N-values. In total, there were 789 samples tested. Of these, 62 were PCR-positive, 52 with the NP Ag RDT, and 44 with the saliva Ag RDT. The sensitivity estimates can therefore never be estimated with the numbers I am seeing in this table. I would provide the actual numbers from which the sensitivity estimates were estimates in the parentheses instead of one N number, for example: 79.2% (XX/XX)

Correction and comment: Numbers were indeed confusing therefore table was edited to make it clearer. 

“Analysis by days post onset did not result in higher sensitivities”. How was this concluded? Looking at table 2, I see percentages for the group 0-3 days post onset and the group 0-7 post onset. For a proper comparison, one would have to make two different groups, for example comparing samples taken 0-3 days after symptom onset, and samples taken 4-7 days post symptom onset.

Correction and comment: in previous studies big difference was usually seen between people who had symptom onset 0-3 days vs 0-7. In this population/study we did not see this due to the uneven split of the population who participated in the study and the comment refers to this fact. It was made clear both by the presented numbers in Table 2 (48 vs 38 people with symptom onset 0-7 days vs 0-3 days) and in the text (in abstract and Line 151-153)

Minor points:

Line 47: People were either tested because of respiratory symptoms or because of being a contact of an individual with a confirmed SARS-CoV-2 infection as mentioned in line 47/48. Could you provide information on the number of samples that were from contact persons, perhaps in Table 1?

Answer: this specifically was not recorded in the questionnaire, one can assume that many of the asymptomatic people were contacts.

Line 62: This sentence requires rewriting. 

Comment: sentence was rephrased (Line 83-85).

Line 98: To me, this summary statistic is redundant and without meaning. 

Comment: whole result section was rewritten and restructured.

Line 99: Sensitivity increased to 88.6, but where is it increasing from?

Comment: sentence was rephrased. Increase from 63%, the overall sensitivity.

---

## [Decision Letter · Decision Letter 1]

19 Nov 2021

Clinical evaluation of the SD Biosensor SARS-CoV-2 saliva antigen rapid test with symptomatic and asymptomatic, non-hospitalized patients.

PONE-D-21-21141R1

Dear Dr. Iglói,

We’re pleased to inform you that your manuscript has been judged scientifically suitable for publication and will be formally accepted for publication once it meets all outstanding technical requirements.

Kind regards,

Etsuro Ito

Academic Editor

PLOS ONE

Reviewers' comments:

Reviewer's Responses to Questions

**Comments to the Author**

1. If the authors have adequately addressed your comments raised in a previous round of review and you feel that this manuscript is now acceptable for publication, you may indicate that here to bypass the “Comments to the Author” section, enter your conflict of interest statement in the “Confidential to Editor” section, and submit your "Accept" recommendation.

Reviewer #1: All comments have been addressed

Reviewer #2: All comments have been addressed

2. Is the manuscript technically sound, and do the data support the conclusions?

Reviewer #1: Yes

Reviewer #2: Yes

3. Has the statistical analysis been performed appropriately and rigorously? 

Reviewer #1: Yes

Reviewer #2: Yes

4. Have the authors made all data underlying the findings in their manuscript fully available?

Reviewer #1: Yes

Reviewer #2: Yes

5. Is the manuscript presented in an intelligible fashion and written in standard English?

Reviewer #1: Yes

Reviewer #2: Yes

6. Review Comments to the Author

Reviewer #1: The edited manuscript successfully addresses all points in the review. I would consider this manuscript appropriately suitable for publication.

Reviewer #2: Excellent revision. Results are very clear.

One final comment from my side, and I leave this up to the authors whether they'd like to change this. I'd change the first sentence of the discussion. I think it should be made in clear what the performance refers to. Is its poorer performance in detecting PCR+ve samples or positive cell cultures?

7. PLOS authors have the option to publish the peer review history of their article (what does this mean?). If published, this will include your full peer review and any attached files.

Reviewer #1: **Yes: **Arnoldus Bosman

Reviewer #2: **Yes: **Tom Woudenberg

---

## [Editor Report · Acceptance letter]

13 Dec 2021

PONE-D-21-21141R1 

Clinical evaluation of the SD Biosensor SARS-CoV-2 saliva antigen rapid test with symptomatic and asymptomatic, non-hospitalized patients. 

Dear Dr. Iglói:

I'm pleased to inform you that your manuscript has been deemed suitable for publication in PLOS ONE. Congratulations! Your manuscript is now with our production department. 

Kind regards, 

on behalf of

Prof. Etsuro Ito 

Academic Editor

PLOS ONE